# Genome-Wide Identification, Phylogenetic Evolution, and Abiotic Stress Response Analyses of the Late Embryogenesis Abundant Gene Family in the Alpine Cold-Tolerant Medicinal *Notopterygium* Species

**DOI:** 10.3390/ijms26020519

**Published:** 2025-01-09

**Authors:** Xuanye Wu, Xiaojing He, Xiaoling Wang, Puyuan Liu, Shaoheng Ai, Xiumeng Liu, Zhonghu Li, Xiaojuan Wang

**Affiliations:** Key Laboratory of Resource Biology and Biotechnology in Western China, Ministry of Education, College of Life Sciences, Northwest University, Xi’an 710069, China; 202221372@stumail.nwu.edu.cn (X.W.); heexj13@163.com (X.H.); wxiaoling98@163.com (X.W.); liupuyuan111@stumail.nwu.edu.cn (P.L.); 17329448694@163.com (S.A.); xiumengliu@163.com (X.L.)

**Keywords:** *Notopterygium*, *LEA* gene family, genome-wide analysis, molecular evolution, expression analysis

## Abstract

Late embryogenesis abundant (LEA) proteins are a class of proteins associated with osmotic regulation and plant tolerance to abiotic stress. However, studies on the *LEA* gene family in the alpine cold-tolerant herb are still limited, and the phylogenetic evolution and biological functions of its family members remain unclear. In this study, we conducted genome-wide identification, phylogenetic evolution, and abiotic stress response analyses of *LEA* family genes in *Notopterygium* species, alpine cold-tolerant medicinal herbs in the Qinghai–Tibet Plateau and adjacent regions. The gene family identification analysis showed that 23, 20, and 20 *LEA* genes were identified in three *Notopterygium* species, *N. franchetii*, *N. incisum*, and *N. forrestii*, respectively. All of these genes can be classified into six LEA subfamilies: LEA_1, LEA_2, LEA_5, LEA_6, DHN (Dehydrin), and SMP (seed maturation protein). The LEA proteins in the three *Notopterygium* species exhibited significant variations in the number of amino acids, physical and chemical properties, subcellular localization, and secondary structure characteristics, primarily demonstrating high hydrophilicity, different stability, and specific subcellular distribution patterns. Meanwhile, we found that the members of the same LEA subfamily shared similar exon–intron structures and conserved motifs. Interestingly, the chromosome distributions of *LEA* genes in *Notopterygium* species were scattered. The results of the collinearity analysis indicate that the expansion of the *LEA* gene family is primarily driven by gene duplication. A *Ka/Ks* analysis showed that paralogous gene pairs were under negative selection in *Notopterygium* species. A promoter *cis*-acting element analysis showed that most *LEA* genes possessed multiple *cis*-elements connected to plant growth and development, stress response, and plant hormone signal transduction. An expression pattern analysis demonstrated the species-specific and tissue-specific expression of *NinLEAs*. Experiments on abiotic stress responses indicated that the *NinLEAs* play a crucial role in the response to high-temperature and drought stresses in *N. franchetii* leaves and roots. These results provide novel insights for further understanding the functions of the *LEA* gene family in the alpine cold-tolerant *Notopterygium* species and also offer a scientific basis for in-depth research on the abiotic stress response mechanisms and stress-resistant breeding.

## 1. Introduction

Plants are subjected to an array of extraneous environments that exert influences on their ontogeny, development, and maturation. In order to accommodate complex and volatile environment conditions, plants have evolved numerous gene families possessing analogous architectures and functionalities. The late embryogenesis abundant protein (LEA) gene family is responsible for encoding LEA proteins [1]. LEA proteins constitute a diverse family of small, hydrophilic polypeptides (10–30 kD); their initial discovery was made during the terminal phases of embryonic development in cotton (*Gossypium hirsutum* L.) seeds [2]. LEA proteins are highly hydrophilic and thermally stable, with their higher-order structures comprising non-periodic linear and α-helix conformations [3], and present in different tissues of plants, including seeds, roots, stems, and buds [4]. Under abiotic stress conditions such as drought, salinity, and extreme temperatures, many plants accumulate highly hydrophilic LEA proteins [5,6] to stabilize vastly different target biomolecules, such as enzymes, membranes, and DNA/RNA, protecting plant tissues from various abiotic stresses [7,8].

Based on sequence characteristics and structural domains, the *LEA* gene family is generally categorized into eight subfamilies: LEA_1, LEA_2, LEA_3, LEA_4, LEA_5, LEA_6, Dehydrin (DHN), and seed maturation protein (SMP) [9]. *LEA* genes have been identified in many plant species, for instance, 51 *AtLEA* were identified in the model plant *Arabidopsis thaliana* (L.) Heynh., and most of them had abscisic acid response elements (ABREs) and temperature response elements (LTREs) in their promoters; many genes containing the respective promoter elements were induced by abscisic acid, temperature, or drought [5]. Meanwhile, an evolutionary analysis showed that the genome of *Oryza sativa* L. and *Triticum aestivum* L. contained 34 and 281 *LEA* family genes [10,11]. A total of 123 *NtLEA* genes were also identified in tobacco (*Nicotiana tabacum* L.), which were significantly induced under NaCl stress [12]. Additionally, some studies have shown that there were significant differences in *LEA* gene expression among species of the same genus with different ploidy levels. Some other studies have conclusively shown that LEA proteins played a pivotal role in plants’ adaptive responses to abiotic stressors, particularly those associated with temperature and water. For example, a study showed that *OsEm1* (LEA protein family) in rice could enhance the tolerance of species to drought stress [13]. In potatoes, ABA induction leads to high expressions of *StLEA1-3* and *StDHN-1*, while *StLEA3-1*, *StDHN-1*, *StDHN-3*, and *StASR-4* are significantly induced by salt stress [12]. The transformation of tobacco with the key gene *PgLEA2-50* enhanced osmoregulation and antioxidant levels in transgenic lines, significantly improving their resistance to abiotic stress [14]. However, the mechanisms of abiotic stress responses of LEA family genes in alpine cold-tolerant herb plants remain unclear.

*Notopterygium*, a genus endemic to China, which belongs to the Apiaceae and is primarily distributed in high-altitude mountainous regions of the Qinghai–Tibet Plateau and its adjacent provinces, such as Shaanxi, Qinghai, Gansu, and Sichuan. This genus encompasses four species clades exhibiting distinct morphological variations, *Notopterygium franchetii*, *Notopterygium incisum*, *Notopterygium oviforme*, and *Notopterygium forrestii* [15], among which the first two are particularly recognized as essential perennial medicinal herbs in China. According to the Chinese Pharmacopoeia, the rhizomes and roots of these herbs are primarily utilized in the treatment of ailments such as wind-cold syndromes with exterior manifestations, headache accompanied by stiffness of the neck, and rheumatic paralysis. However, due to human exploitation and destruction, coupled with the impacts of global climate change on their unique natural habitats, wild resources are scarce, and there is an urgent need to develop resistant varieties. The identification of resistant genes and/or family genes is very important to develop novel resources or varieties. In a previous study, Jia et al. (2017) conducted transcriptome sequencing on *N. incisum* and *N. franchetii* to explore environmental adaptation genes, and their research revealed the presence of various environmental stress genes regulating substance and energy metabolism in both species [16]. In a subsequent study by Jia et al. (2019), transcriptome sequencing was performed on the flowers, leaves, and stems of *N. incisum*, and 21 genes significantly associated with cold tolerance were identified [17]. Currently, there remains a scarcity of research on the molecular mechanisms of abiotic stress responses in *Notopterygium* species. It is strongly necessary to conduct a genome-wide identification of resistant family genes in *Notopterygium* species and to further investigate the evolutionary history of this gene family and its relationship with the diversity in the sequence and function of LEA proteins. In this study, we conducted the first genome-wide analysis of *LEA* genes in *Notopterygium* species, aiming to characterize their sequences, evolutionary relationship, putative function, and expression profiles in response to drought and temperature stresses. Our research provides invaluable insights that will pave the way for the future exploration of *LEA* genes in the alpine cold-tolerant *Notopterygium* species.

## 2. Results

### 2.1. The Identification of the LEA Genes in Notopterygium

A total of 20, 20, and 23 *LEA* gens were identified in *N. franchetii* (*NfrLEA1*-*NfrLEA23*), *N. forrestii* (*NfoLEA1*-*NfoLEA23*), and *N. incisum (NinLEA1*-*NinLEA23*), which were classified into six subfamilies, LEA_1, LEA_2, LEA_5, LEA_6, DHN, and SMP, based on protein conserved domains and a phylogenetic analysis. SMP and LEA_5 are the top two subfamilies with the largest numbers of members (Figure 1 and Appendix A).

The protein lengths of coding sequences range from 84 (*NfrLEA4* and *NfoLEA16*) to 455 amino acids (*NfrLEA11* and *NfrLEA15*) with an average sequence length of 218 amino acids. The theoretical molecular weight of LEA proteins ranges from 9.05 to 50.33 kiloDaltons (kDs), and the theoretical isoelectric point (pI) values of 15 out of 23 LEA proteins in *N. franchetii*, 13 out of 20 LEA proteins in *N. forrestii*, and 13 out of 20 LEA proteins in *N. incisum* are less than 7.0, indicating that most of the LEA proteins in *Notopterygium* species are acid proteins. The instability indices range from 8.15 to 63.36, and LEA proteins in the LEA_5 subfamily are unstable proteins. The aliphatic index ranges from 24.3 to 110.68, with 94% of LEA proteins having an aliphatic index below 100 and a grand average of hydropathicity (GRAVY) value less than 0, indicating high hydrophilicity (ranging from −1.385 to −0.062). Subcellular localization prediction indicated that LEA proteins are mainly distributed in the nucleus, particularly those belonging to the DHN subfamily and the LEA_6 subfamily, while the remaining LEA proteins are distributed in the nucleus, cytoplasm, mitochondrion, and plasma membrane (Appendix A). The results of secondary structure predictions for the amino acid sequences encoded by *LEA* genes are shown in Appendix A. Among the amino acid sequences, the secondary structure is predominantly a random coil structure with a water content of more than 20%, ranging from 24.66% to 79.76%, while β-turn has the lowest proportion, ranging from 0.00% to 17.7%. The secondary structure of LEA proteins in the LEA_2 subfamily is relatively regular, and it has the highest proportion of extended strands among the six subfamilies.

### 2.2. Phylogenetic Analysis and Classification of LEA Gene Family

To determine the classification and evolutionary relationships of the identified *LEA* gene family, a phylogenetic tree was constructed based on the multiple sequence alignment of 227 LEA protein sequences, including 23 in *N. franchetii*, 20 in *N. forrestii*, 20 in *N. incisum*, 51 in *A. thaliana*, 35 in *C. sativum*, 27 in *A. graveolens*, 31 in *D. carota*, 20 in *A. sinensis*, 31 in *P. ginseng*, and 17 in *Panax notoginseng* (Figure 1). According to the constructed phylogenetic relationship, the 227 LEA proteins were classified into eight major phylogenetic groups: LEA_1, LEA_2, LEA_3, LEA_4, LEA_5, LEA_6, DHN, and SMP. Of these, DHN and SMP are the largest groups with 60 and 58 *LEA* gene family members, respectively, and LEA_6 is the least abundant member of the subfamily with 10 members. The distribution ratio of *LEA* gene family members among various subfamilies follows this pattern consistently across every species. In the LEA_4 subfamily, only four species, *A. sinensis*, *P. ginseng*, *P. notoginseng*, and *A. thaliana*, contain members of the *LEA* gene family, while no other species do. This suggests that gene loss events have occurred during the evolution of LEA_4 subfamily members. Furthermore, *AthLEA21* and *AthLEA22* do not belong to any subfamily, indicating that these two genes exhibit significant sequence divergence and are phylogenetically distant from *LEA* gene family members in other species. In the earliest classification of the *A. thaliana LEA* gene family, these two genes were assigned to the AtM subfamily [5]. No AtM members were found in the *Notopterygium* species and the other six species examined, suggesting that these two genes may have undergone gene loss events during evolution. Additionally, it was observed that individual members of other subfamilies appear in the branches of each subfamily. This could be due to the duplication and loss of a small portion of *LEA* genes during evolution, accompanied by gene neofunctionalization and subfunctionalization. Paralogous genes of various species show considerable divergence, ultimately leading to low sequence similarity among some *LEA* gene family members across different species.

### 2.3. Gene Structure, Protein Conserved Motif, and Cis-Acting Element Analysis

The exon–intron structures of all the identified *LEA* genes were investigated to better understand the structural diversity of these genes in *Notopterygium* species. The results reveal (Figure 2D) that the majority of members possess 1–2 introns, with the SMP subfamily exhibiting the highest number of introns among *LEA* genes, where each member contains at least two introns. There are some differences in the number of exons and the length of introns among members of the *LEA* gene family. For example, two members of the LEA_6 subfamily lack introns, while members of the LEA_1, LEA_2, LEA_5, and DHN subfamilies predominantly contain a single intron. Furthermore, it was observed that, except for *NinLEA18*, which has a unique structure characterized by a notably long intron segment, the intron–exon structural features of *LEA* gene family members belonging to the same subfamily are highly similar. These findings suggest that the phylogenetic relationships and subfamily classifications in the *LEA* gene family are closely associated with their gene structures.

To further reveal the structural and functional characteristics of LEA proteins, the online website MEME was employed to analyze the conserved structural motifs, and detailed motif information is provided in Appendix A. A total of 15 conserved motifs were identified and named motif 1 to motif 15. As shown in Figure 2B, no member has a complete set of 15 conserved motifs, and the motif numbers of the LEA family members range from one to seven. Sequences grouped in the same subfamily share an identical number of conserved motifs and types of motifs. Notably, the LEA_6 subfamily lacks any conserved motifs and exhibits a short sequence length, suggesting a potential loss of conserved motifs during evolutionary processes for members of this subfamily. In contrast, the LEA_1 subfamily displays a lower number and variety of conserved motifs in their protein sequences, with some members possessing only one conserved motif. All members of this subfamily contain motif 14, and three members also harbor motif 4, so they were grouped into a distinct category. LEA members of the DHN subfamily consistently contain motif 7 and motif 11, while some members additionally possess motif 12, and others have motifs 14 and 5. These findings suggest that *LEA* genes in the same subfamily exhibit higher homology and closer evolutionary relationships.

In addition, 12 putative conserved domains were identified and inhibited, as shown in Figure 2C. In general, the 63 conserved domains of LEA proteins in *Notopterygium* species lack high similarity, but unique conserved segments exist in each *LEA* gene subfamily. Specifically, the SMP subfamily contains the SMP domain, and some members also include the ACD_sHsps_p23-like domain. The DHN subfamily is characterized by the DHN domain. The domains of LEA_2 subfamily members are divided into two categories: those containing only the LEA_2 superfamily and those containing both the LEA_2 and Why domains. Members of the LEA_5 subfamily are classified into those that contain either the LEA_5 or LEA_5 superfamily domain. Members of the LEA_1 and LEA_6 subfamilies each contain only a single domain, the LEA_1 or LEA_6 domain, respectively. These results reveal that members in the *LEA* gene family that are phylogenetically close have similar compositions of conserved domain and similar functions. The conserved domains and motifs of LEA proteins in the same subfamily are highly consistent, while there are significant differences between different subfamilies.

An analysis of the 2000 bp sequence upstream of the transcription start site of *LEAs* in *Notopterygium* identified 1718 putative *cis*-elements related to stress response (Figure 3 and Appendix A). The promoter region of the LEA gene family mainly contains responsive elements such as light-regulated elements, abscisic acid (ABA), light, methyl jasmonate (MeJA), and anaerobic induction. Hormone-responsive elements were found in the promoter regions of most *LEA* genes. Among them, 84% of the *LEA* genes contain ABA-responsive elements, participating in the regulation of the ABA signaling pathway involved in abiotic stress responses. Additionally, 67.6% of *LEA* gene promoters harbor MeJA-responsive elements. Furthermore, auxin-, gibberellin-, and salicylic acid-responsive elements were also detected, all of which are associated with plant stress resistance processes. Simultaneously, a large number of *cis*-elements directly related to drought, anaerobiosis, low temperature, defense, and salt stresses were identified. In summary, *LEA* genes may participate in *Notopterygium* species growth and development through multiple pathways and respond to various abiotic stresses.

### 2.4. PPI Networks and GO Annotation

The PPI network of LEA proteins is shown in Figure 4A; 11 LEA proteins, namely NinLEA2, NinLEA6, NinLEA7, NinLEA12, NinLEA17, NinLEA18, NinLEA19, NinLEA20, NfoLEA16, NfoLEA18, and NfrLEA19, were mapped onto the PPI network, and 6 LEA proteins were significantly interconnected among them. Specifically, NinLEA6 and NinLEA17 function as molecular chaperones, stabilizing protein structures and protecting proteins and membranes from aggregation, which is crucial for stress tolerance, particularly against dehydration and low-temperature stress. NinLEA12 is involved in dehydration tolerance, while NinLEA19 enhances plant tolerance to osmotic stress during germination and seedling stages. Additionally, strong interactions were observed between NinLEA6 and NinLEA12, between NinLEA6 and NinLEA19, as well as between NinLEA19 and NinLEA20, indicating that these proteins share similar functions and participate in biological processes related to plant responses to external stress through their interactions. The remaining five LEA proteins were relatively isolated, and their protein functions remain unclear.

The GO annotation results reveal that *LEA* genes are broadly involved in annotations related to biological processes, molecular functions, and cellular components (Figure 4B–D). In biological processes, they are all implicated in key processes such as responses to chemicals, growth and development, and seed and fruit development. Additionally, binding functions in molecular functions and cellular components such as cytosol and cytoplasm are also common annotation characteristics shared by these three species. These commonalities indicate a certain degree of conservation in *LEA* genes in terms of basic life activities and cellular constitution. A total of 11 *NfrLEA* genes were mapped to different GO functional nodes, with a relatively large proportion of annotations related to metabolic processes, cellular processes, and growth and development, suggesting their specific functions in these aspects (Figure 4B). Fourteen mapped *NfoLEA* genes exhibited prominent responses to chemicals such as ABA oxygenated compounds, which may be relevant to their hormone regulation and antioxidant functions (Figure 4C). Thirteen *NinLEA* genes were mapped and are associated with osmotic regulation, demonstrating their unique functions in resisting abiotic stress (Figure 4D). The differences in the GO annotation results reflect the specific physiological strategies developed by the *Notopterygium* species when adapting to their environment.

### 2.5. Chromosomal Distribution, Collinearity, and Selective Pressure Analysis

Twenty-one *NfrLEA* genes were precisely mapped to eight chromosomes of *N. franchetii*. Chromosome 1 (Chr1) harbored the largest number of genes, with six *NfrLEA* genes concentrated at the chromosomal ends (Figure 5). Five *NfrLEA* genes were distributed on Chr3, and most of them on chromosomes were located at their ends. All 20 *NfoLEA* genes were also specifically localized to chromosomes, with Chr10 having the highest number of genes at six, clustered together. All 20 *NinLEA* genes were precisely mapped to ten chromosomes, with 3 *NinLEA* genes each on Chr1, Chr2, Chr4, and Chr10. Overall, *NinLEA* genes were more dispersed across the chromosomes. Notably, no *LEA* genes from three species were found on Chr7.

Eighteen and twenty-four pairs of paralogous genes encompassed all members of *NfrLEAs and NfoLEAs* (Figure 6A,B). In *N. incisum*, the *LEA* gene family comprised approximately 50 pairs of homologous *NinLEAs* (Figure 6C), double the number found in both *N. franchetii* and *N. forrestii*. To explore potential evolutionary clues of the *LEA* gene family, an interspecific collinearity graph of *Notopterygium* associated with model plant *A. thaliana* was constructed (Figure 6D). The results indicate that there are 39 and 35 orthologous pairs between *N. franchetii* and *N. incisum* and between *N. incisum* and *N. forrestii*, respectively, and collinearity was mainly distributed on Chr1, Chr2, Chr4, Chr9, and Chr 10. Only 11 pairs of LEA genes showed collinearity between *A. thaliana* and *N. franchetii*. These findings suggest that *LEA* gene families in *Notopterygium* species are more closely related and may share a common ancestor and that *LEA* genes may not originate from *A. thaliana*.

The *Ka/Ks* values were calculated to investigate potential selective pressure during the evolution of the *LEA* gene family in *Notopterygium* species. The results show that the *Ka/Ks* values of all gene pairs exhibit *Ka/Ks* < 1 (Appendix A), suggesting that the repetitive *LEA* genes are primarily constrained by intense purification selection pressure in *Notopterygium* species.

### 2.6. Expression Pattern Analyses of NinLEA in Notopterygium Species

To reveal the tissue-specific and species-specific expression of *LEA* genes, this study conducted a transcriptome analysis on different tissues (roots, stems, leaves, flowers, and fruits) of four *Notopterygium* species (Figure 7). The results indicate that *NinLEA10, NinLEA11*, and *NinLEA13* were not expressed in any of the four species. Among the *NinLEAs* with detectable expression levels, most were expressed at different levels in different tissues and species, and the expression of some genes was highly tissue-specific and species-specific. In *N. franchetii*, *NinLEA2*, *NinLEA3*, and *NinLEA6* were significantly more expressed in root tissues than in other tissues, while *NinLEA1*, *NinLEA5*, *NinLEA7*, *NinLEA9*, *NinLEA18*, and *NinLEA20* had the highest expression levels in flowers, and *NinLEA4*, *NinLEA15*, *NinLEA16*, and *NinLEA19* had the highest expression levels in leaves. In *N. incisum* and *N. oviforme*, *NinLEAs* were mainly highly expressed in the roots and flowers, with lower expression being found in the leaves (Figure 7B,D). Different *NinLEAs* were highly expressed in various tissues of *N. forrestii* (Figure 7C). These results indicate that the tissue-specific expression of *NinLEAs* exists and differs among *Notopterygium* species. Additionally, *NinLEA14* and *NinLEA17* were only expressed in *N. forrestii*, but it was also the only species in which *NinLEA20* was not expressed (Figure 7C), while *NinLEA8* was only not expressed in *N.franchetii* (Figure 7A). Notably, *NinLEA2* and *NinLEA3* had relatively high expression levels in the roots of all species, suggesting that they may be essential for root development in *Notopterygium* species.

### 2.7. Expression Analysis of NinLEAs in N. franchetii Under Different Abiotic Stress

Based on a transcriptome expression pattern clustering analysis, six genes (*NinLEA1*, *NinLEA2*, *NinLEA3*, *NinLEA9*, *NinLEA19*, and *NinLEA20*) that exhibit stable expression across different tissues and species yet demonstrate high specificity in *N. franchetii* were selected. The responses of these genes to high-temperature, low-temperature, and drought stresses were analyzed using qRT-PCR. The results indicate that the relative expression levels of the *NinLEAs* underwent significant changes under three stress treatments, suggesting their involvement in the response of *N. franchetii* to high-temperature, low-temperature, and drought stresses (Figure 8). Furthermore, variations in the relative expression levels of different *NinLEAs* were observed. For instance, under high-temperature stress, the relative expression of *NinLEA19* in leaves continuously increased over a period of 0–12 h, while the expression trends of the other genes first rose, then declined, and subsequently rose again within the same time frame. In roots, the relative expression levels of *NinLEA1*, *NinLEA2*, *NinLEA3*, and *NinLEA19* first decreased and then increased, whereas those of *NinLEA9* and *NinLEA20* exhibited the opposite trend (Figure 8A). Under continuous low-temperature conditions, the relative expression levels of *NinLEA1*, *NinLEA3*, and *NinLEA20* in the leaves exhibited a trend of initially increasing and then decreasing (Figure 8C). We also found that *NinLEA3* exhibits a similar expression pattern under both low- and high-temperature stresses. Additionally, the relative expression levels of the same gene in different tissues varied under the same stress. For example, under drought stress, the relative expression of *NinLEA3* in the leaves first increased and then decreased, while in the roots, it first decreased and then increased, showing the complete opposite trend. Under high-temperature stress, the changes in all *NinLEA*s were more pronounced in the leaves compared to the roots (Figure 8B).

## 3. Discussion

The *LEA* gene family occupies a central stage in both the intricate process of plant embryonic development and the vital mechanisms underlying plant resilience to abiotic stressors such as drought and cold [18]. The proteins encoded by *LEA* genes, characterized as multifaceted functional proteins in plants, exhibit remarkable capabilities in scavenging intracellular ROS and safeguarding macromolecular structures, thereby effectively mitigating the detrimental impacts of abiotic stresses on plant organisms [19]. *Notopterygium* species are currently confined to the ecologically unique Qinghai–Tibet Plateau and its adjacent high-altitude mountainous habitats [20], where they thrive under exacting environmental conditions and are particularly susceptible to a variety of influences from abiotic stresses. Notably, *N. franchetii* and *N. incisum* are esteemed medicinal herbs of considerable significance, and the meticulous identification of genes associated with their remarkable stress resistance holds profound scientific and practical value.

In this study, we identified a total of 63 *LEA* genes belonging to six LEA subfamilies from the genomes of three *Notopterygium* species (23 *NfrLEAs*, 20 *NinLEAs*, and 20 *NfoLEAs*), with a member count comparable to that observed in tomato, rice, tea plant, and other plants such as *A. graveolens*, *A. sinensis*, and *P. notoginseng* (Figure 1). However, in species such as tobacco, cotton, and ginseng, the number of LEA gene family members exceeds 100 [21,22,23]. This may be attributed to the different selective pressures and genetic variations in these species during evolution, leading to the expansion of the *LEA* gene family and the accumulation of more *LEA* genes to enhance stress resistance. Among the six LEA subfamilies, the SMP subfamily dominates with 18 members. The LEA_3 and LEA_4 subfamilies were not observed (Figure 1). Similarly, members of the LEA_4 subfamily were also absent in the *LEA* gene families of Apiaceae species, such as *C. sativum*, *A. graveolens*, and *D. carota* in this study, and the number of LEA_3 subfamily members was extremely low (Figure 1). This suggests that a significant gene loss may have occurred for LEA_3 and LEA_4 subfamily members during the evolution of Apiaceae plants. Variations in the number and types of *LEA* genes exist among different species. For example, the LEA4 subfamily dominates in *Brassica campestris* [5], while in *Prunus mume* and wheat, the DHN subfamily has the largest number of members [24,25]. In tobacco and ginseng, the LEA_2 subfamily has the highest number of members [12,14], indicating the complexity of LEA gene evolution.

Subcellular localization prediction indicates that LEA proteins are primarily distributed in the nucleus, cytoplasm, and mitochondria, with a smaller proportion located in the plasma membrane. These findings align with previous studies on *A. thaliana* and *Pterocarya stenoptera* [5,26], reflecting the diversity of intracellular LEA protein functions and mechanisms in *Notopterygium* species (Appendix A). An analysis of physical and chemical properties revealed that, except for the LEA_5 subfamily, the proteins encoded by *LEA* genes from the other subfamilies are stable. Furthermore, 94% of LEA proteins exhibit high hydrophilicity. LEA proteins with high hydrophilicity and thermostability can redirect water molecules, bind salt ions, and subsequently eliminate reactive oxygen species (ROS) that accumulate in cells due to external stress stimulation [27]. These properties contribute to the tolerance to dehydration and other stresses in plants [9].

In this study, members of the same subfamily are mostly clustered in the same branch, indicating a high degree of evolutionary conservation across species (Figure 1). The intron–exon structural characteristics of *LEA* gene family members belonging to the same subfamily are highly similar. A total of 89% of LEA gene family members have 0–2 introns. Members of the LEA_6 subfamily lack introns, while members of the LEA_1, LEA_2, LEA_5, and DHN subfamilies predominantly contain only one intron (Figure 2D). Previous studies have found that genes related to stress responses generally have a lower number of introns, and genes with fewer introns are beneficial for reducing transcription costs and enabling faster transcription and expression processes, thereby allowing cells to rapidly respond to abiotic stresses [28]. For example, most *LEA* genes in *A. thaliana* contain only one intron [5], and more than half of the *LEA* genes in wheat lack introns [10].

During the course of prolonged evolution, plants have developed a set of complex strategies to adapt and survive under harsh environmental conditions, with transcriptional changes being a primary adaptive mechanism. *Cis*-acting elements on gene promoters regulate the mRNA abundance of numerous downstream target genes through interactions with transcription factors, playing a crucial role in plant responses to abiotic stresses [29]. For instance, the MBS element is associated with drought stress responses [30], while TC-rich repeats are related with salt stress responses [31]. Therefore, analyzing the *cis*-acting elements in gene promoter regions aids in identifying specific functions of genes. In this study, 53 *LEA* genes contained ABA-responsive elements, 67.6% of LEA gene promoters harbored MeJA-responsive elements, and 91.6% of LEA genes contained MYB and light-responsive elements. Studies on wheat, tea plants, and ginseng have found similar regulatory elements in the upstream regulatory regions of *LEA* genes [14,23,25]. Additionally, a large number of elements are involved in responses to other compounds and non-ABA-dependent stress responses. These elements may directly or indirectly participate in stress responses, regulate growth and development, and maintain cellular homeostasis, enabling *LEA* genes to respond to abiotic stresses through multiple pathways. For *Notopterygium* species, the same gene may play a role in different tissues, for example, a gene is expressed in the root, flower, and leaf of *N. franchetii* (Appendix A); three genes are expressed in the root, flower, and leaf of *N. incisum* (Appendix A); one gene is expressed in various tissues of *N. forbesii* (Appendix A); three genes are expressed in various tissues of *N. oviforme* (Appendix A). Meanwhile, members of the same gene family may also be expressed in the same tissues across different species (Appendix A).

LEA proteins can prevent the collapse of cellular structures by binding to cell membranes [32], and they can act as molecular chaperones by binding to misfolded proteins and repairing misassembled proteins to restore their biological activity [33]. The PPI network analysis conducted in this study revealed that NinLEA6 and NinLEA17 proteins function as molecular chaperones to stabilize protein structures (Figure 4A). Both *NinLEA6* and *NinLEA17* genes belong to the LEA_5 subfamily, and research on *A. thaliana* has shown that LEA_5 regulates organellar translation to enhance respiration relative to photosynthesis in response to stress [34]. The GO enrichment analysis results indicate that there are partial differences in the functions of *LEA* genes among the three *Notopterygium* species. The functions of *LEA* genes in *N. franchetii* were primarily relevant to metabolic processes and growth and development, while those in *N. forrestii* were mainly involved in hormone regulation and antioxidant activities. The *LEA* genes in *N. incisum* were primarily associated with osmoregulation (Figure 4B–D). However, these biological processes were all related to the function of combating abiotic stresses, confirming the results of the *cis*-acting element analysis in the promoter regions, which indicate that the pathways through which *LEA* genes in *Notopterygium* species respond to abiotic stresses are not singular.

In this study, the results of chromosomal distribution show that most members of the *LEA* gene family in each species are scattered at different locations on different chromosomes in the three *Notopterygium* species (Figure 5). The formation of this phenomenon is significantly related to chromosome segment duplication and tandem duplication events, which have led to changes in the chromosomal locations of gene family members [35,36]. The dispersed distribution of genes may indicate that these genes have undergone different selective pressures during evolution, resulting in functional differentiation. Alternatively, it may increase the complexity of gene expression regulation, where different gene family members may be controlled by distinct regulatory mechanisms to achieve specific temporal and spatial expression patterns [37]. This regulatory complexity aids organisms in achieving precise gene expression regulation during growth and development. The *Ka/Ks* values for paralogous gene pairs were less than 1 (Appendix A), indicating that the LEA gene family members in the three *Notopterygium* species underwent rigorous purifying selection during their evolution and expansion. Similarly, strong purifying selection effects have been observed in the evolution of *LEA* gene families in other species, such as *Populus trichocarpa*, *Triticum aestivum*, and maize [25,38,39]. The *NfrLEAs* exhibited higher homology with *NfoLEAs* and *NinLEAs*, with similar numbers of orthologous gene pairs, but showed the lowest homology with *AtLEAs*. This suggests that there is a closer genetic relationship among the *LEA* gene families in *Notopterygium* species.

Through the analysis of RNA-Seq data, we found that not all *NinLEAs* are expressed in the tissues of the three *Notopterygium* species, and there are differences in expression abundance and tissue specificity among the co-expressed *NinLEAs* (Figure 7). In most species, *LEA* genes exhibit distinct tissue-specific expression patterns. For example, in *N. forresii*, *NinLEA2*, *NinLEA3*, and *NinLEA6* are highly expressed in root tissues, while *NinLEA1*, *NinLEA5*, and others are predominantly expressed in flowers, and *NinLEA4* is abundant in the leaves (Figure 7A). In *N. incisum*, most of the expressed *LEA* genes have the highest expression levels in the roots, whereas *NinLEA4* and *NinLEA7* are highly expressed in the flowers (Figure 7B). Similar tissue-specific expression patterns were observed in *N. forrestii* and *N. oviforme*, although specific genes and tissues with high expression differed. Across species, there were variations in the expression patterns of *LEA* genes. Genes such as *NinLEA5*, *NinLEA10*, and *NinLEA11* were not expressed in *N. forrestii* and *N. oviforme* but showed different expression profiles in *N. franchetii* and *N. incisum*. The same gene may also have different tissues with high expression in different species. For instance, *NinLEA3* was highly expressed in flowers in both *N. franchetii* and *N. forrestii* but was abundant in the roots in *N. incisum* (Figure 7). Additionally, *NinLEA8* and *NinLEA10* were not expressed or were expressed at extremely low levels in multiple tissues across multiple species. The *LEA* genes in the four *Notopterygium* species exhibited complex expression patterns and characteristics in different tissues, which may reflect their different functions and roles in plant growth and development and in the response to environmental stress. In contrast, most *NnLEA* genes in lotus are primarily expressed during the later stages of cotyledon and plumule development, indicating their crucial roles in seed maturation [40]. The *LEA* genes in *Arachis hypogaea* usually exhibit relatively consistent expression but at generally low levels, while genes in the LEA2, LEA3, and DHN subfamilies are strongly expressed in the roots, leaves, and flowers [41], which is similar to the expression pattern of *NinLEA* in *N. franchetii*. Most *LEA* genes in *Juglans regia* are expressed in leaf and flower tissues [42], resembling the expression pattern of *NinLEAs* in *N. forrestii*. We further analyzed the responses of *NinLEA* genes to high-temperature, low-temperature, and drought stresses through RNA-Seq data and qRT-PCR validation (Figure 8). The relative expression levels of *NinLEA* genes underwent significant changes after three stress treatments. Compared with high-temperature stress, the changes in the relative expression levels of *NinLEA* genes in leaf tissues under drought stress were relatively insignificant, and the variation ranges of expression levels in both leaf and root tissues were generally similar. Additionally, we found that *NinLEA3* exhibited a similar expression pattern in the leaves under both high- and low-temperature conditions, and the relative expression levels of *LEA* genes showed a significant increase over a period of 0–2 h under low-temperature conditions. These results suggest that *LEA* genes play a significant role in helping *Notopterygium* species cope with extreme environments. *Notopterygium* species grow in high-altitude regions and often face challenges posed by extreme environmental conditions, such as low temperatures and water scarcity. Therefore, identifying the members of the LEA gene family in *Notopterygium* species and understanding their expression patterns under extreme conditions such as low temperatures, high temperatures, and drought will help us understand the response mechanisms of *Notopterygium* species in extreme environments. This study not only provides valuable stress-resistant gene resources in *Notopterygium* species but also provides profound insights into how these species respond to extreme climate events.

## 4. Materials and Methods

### 4.1. Identification and Characterization of LEA Genes in Notopterygium

The whole genome sequencing data of the three species of *Notopterygium* were obtained from the research group. Based on two methods, *LEA* gene family members were identified. Firstly, the hidden Markov models (HMMs) corresponding to the LEA domain (PF03760, PF03168, PF03242, PF02987, PF00477, PF00257, PF04927, and PF10714) were downloaded from the Pfam database v34.0 (http://pfam.sanger.ac.uk/, accessed on 22 March 2024). The LEA protein sequences of *Notopterygium* species were aligned using the HMM model in HMMER3.0 (http://plants.ensembl.org/hmmer/index.html, accessed on 24 March 2024), using an Expect value (E value) of 10^−5^ [43]. Then, the protein sequences of *A. thaliana* were obtained from the TAIR database (https://www.arabidopsis.org/, accessed on 25 March 2024) and were used to perform a BLASTP search in the *Notopterygium* species protein database for proteins exhibiting E values less than 10^−5^. Afterwards, the above two results were intersected, incomplete sequences and redundant sequences were removed, and the NCBI-CDD database (https://www.ncbi.nlm.nih.gov/, accessed on 4 April 2024) was used for verification.

ProtParam (https://web.expasy.org/protparam/, accessed on 7 April 2024) was used to predict the physical and chemical properties. The WoLF PSORT web tool (https://web.expasy.org/protparam/, accessed on 8 April 2024) [44] and the SOPMA (https://npsa-prabi.ibcp.fr/cgi-bin/npsa_automat.pl?page=/NPSA/npsa_sopma.html, accessed on 21 April 2024) [45] were used to predict the subcellular localization and the secondary structure composition of LEA proteins, respectively.

### 4.2. Sequence Alignment and Phylogenetic Analysis

To gain further understanding of the evolutionary relationships among the *LEA* genes, we compared the genes in *Notopterygium*, *A. thalian*, *Coriandrum sativum* L., *Apium graveolens* L., *Daucus carota* L., *Angelica sinensis* (Oliv.) Diels, *ginseng* (*Panax ginseng* C. A. Mey.), and *Panax notoginseng* (Burkill) F. H. Chen ex C. H. Chow. Full-length alignment of the *LEA* gene sequences was performed using MUSCLE [46], and Maximum Likelihood (ML) trees were constructed using the MEGA X software (version 11.0.13) with a bootstrap value of 1000 [47].

### 4.3. Analyses of Gene Structure, Protein Conserved Motifs, and Promoter Cis-Acting Elements 

Conserved motifs were predicted for each LEA protein sequence using the MEME web server v5.33 (https://meme-suite.org/meme/tools/meme, accessed on 24 April 2024), with the maximum number of motifs set to 15 and all other parameters set to default values [48].

The promoter sequences of all *LEA* genes, approximately 2000 bp upstream of the transcription start site (TSS), were extracted from the genome data of *Notopterygium* species. The *cis*-acting elements in the promoter region were predicted using the PlantCARE database (https://bioinformatics.psb.ugent.be/webtools/plantcare/html/, accessed on 28 April 2024) [49]. All of the above results were visualized using TBtools software (version 2.056) [50].

### 4.4. Prediction of Protein–Protein Interaction (PPI) Networks and Gene Ontology (GO) Annotation

STRING (https://string-db.org/, accessed on 25 April 2024) program was used to predict the functional interacting network models of LEA proteins; the required score was set at a threshold of 0.40. The eggNOG-mapper (http://eggnog-mapper.embl.de/, accessed on 25 April 2024) was used to obtain functional annotation information for *LEA* genes [51]. Finally, enrichment analysis results were generated using TBtools (version 2.056) [50].

### 4.5. Chromosomal Distribution, Collinearity, and Selective Pressure Analysis

Genome data for *Arabidopsis* were downloaded from the Ensembl Plants database (http://plants.ensembl.org/info/data/ftp/index.html, accessed on 3 May 2024). The TBtools software (version 2.056) was used to analyze and visualize the chromosomal distribution. The MCScanX software (version 11.0.13) was used to search the gene level collinearity [52]. The *Ka/Ks* Calculator v3.0 tool was used to calculate the ratio of non-synonymous to synonymous substitution (*Ka/Ks)* of duplicated gene pairs [53].

### 4.6. Expression Pattern Analysis

To gain insights into the expression patterns of *LEA* genes, transcriptome datasets of *Notopterygium* were analyzed. The transcriptomic data of *LEA* genes in five tissues (roots, stems, leaves, fruits, and flowers) were obtained by RNA-seq to detect the expression profile. Raw data were filtered using fastp with the *N. incisum* genome serving as the reference. Alignment was performed using Hisat2 (version 2.2.0) [54], and unqualified alignments were removed with SAMtools (version 1.17) [55]. StringTie (version 2.2.3) was used for single-sample assembly and quantification [56]. Differential expression analysis was conducted using edgeR (version 3.32.1) with Trimmed Mean of M-values (TMM) normalization and a Biological Coefficient of Variation (BCV) set to 0.1. Venn diagrams, set plots, and cluster analysis heatmaps were generated using R language.

### 4.7. Plant Materials, Growth Condition, and Abiotic Stress in N. franchetii

Three-year-old *N. franchetii* seedlings, with vigorous growth, similar morphology, and intact root systems, were selected as plant material and placed in an intelligent light-controlled incubator for the following three treatments: (1) high-temperature stress treatment, where cultivation conditions were 40 °C with a 12 h light/dark cycle, and leaf and root samples were collected at 0 h, 2 h, 6 h, 12 h, and 36 h; (2) low-temperature stress treatment, where cultivation conditions were 0 °C with a 12 h light/dark cycle, and leaf samples were collected at 0 h, 2 h, 6 h, 12 h, and 24 h; (3) and drought stress treatment, where cultivation conditions were 15 ± 2 °C with a 12 h light/dark cycle. The plants were treated with 40% PEG solution, and leaf and root samples were collected at 0 h, 2 h, 6 h, 12 h, and 24 h.

### 4.8. RNA Extraction and Quantitative Real-Time PCR

Total RNA was extracted from the collected *N. franchetii* samples using a widely recognized TRIzol kit (Accurate Biology, Changsha, China) following the manufacturer’s instructions. RNA quality was assessed using an ultra-micro spectrophotometer, NanoDrop2000 (Kaiao Technology Development Co., Ltd., Beijing, China), which can detect the integrity and concentration, and the results were confirmed using RNase agarose-free gel electrophoresis. After RNA isolation, cDNA synthesis was performed using Hifair^®^III 1st Strand cDNA Synthesis SuperMix for qRT-PCR (gDNA digester plus) (Yeasen, Shanghai, China). For quantitative real-time PCR (Bioer 9600 FQD-96C, Hangzhou, China), primers were designed using Primer Premier 6.0 software (Premier Biosoft Inc., San Francisco, CA, USA (https://premierbiosoft.com/, accessed 26 April 2024)), and the primer design sequences are shown in Appendix A. The qRT-PCR reaction mixture consisted of 3.6 µL ddH_2_O, 1 µL cDNA, 0.4 µL specific primers, and 5 µL ChamQ SYBR qPCR Master Mix (Vazyme Biotech Co., Ltd., Nanjing, China). The amplicon length of the target gene ranges from 453 bp to 1368 bp. The PCR procedure was as follows: pre-incubation at 95 °C for 5 min, followed by 40 cycles of denaturation at 95 °C for 10 s, and annealing at 60 °C for 30 s, with a melting curve analysis to confirm primer and gene specificity. Each procedure included three biological replications and two technical replications. The relative expression levels of *NinLEA* genes were calculated using the 2^−ΔΔCT^ method [57]. Actin8 from *N. franchetii* was used as internal reference gene. We calculated the stability of *NinLEA* genes and reference gene using Bestkeeper [58].

## 5. Conclusions

In this study, a comprehensive and systematic characterization of the *LEA* gene family in three *Notopterygium* species was conducted. Totals of 23, 20, and 20 *LEA* genes were identified in the genomes of *N. franchetti*, *N. incisum*, and *N. forrestii*, respectively. A phylogenetic analysis revealed that the identified *LEA* genes in *Notopterygium* could be classified into six major groups, with no apparent pattern of distribution on the chromosomes. Members in the same group exhibited similar gene structures and motif compositions. A molecular evolutionary analysis suggested that the *LEA* gene families in *Notopterygium* species evolved under the influence of purifying selection, which eliminated deleterious mutation sites during evolution. An analysis of promoter *cis*-elements, expression patterns, and expression characteristics under abiotic stress conditions indicated that some *NinLEAs* may be involved in the growth and development of *Notopterygium* species as well as their response to high-temperature, low-temperature, and drought stresses. These findings provide theoretical support for further investigation into the functions and mechanisms of *LEA* genes in response to abiotic stress in the alpine cold-tolerant *Notopterygium* species, laying the foundation for molecular breeding in these species.

## Figures and Tables

**Figure 1 ijms-26-00519-f001:**
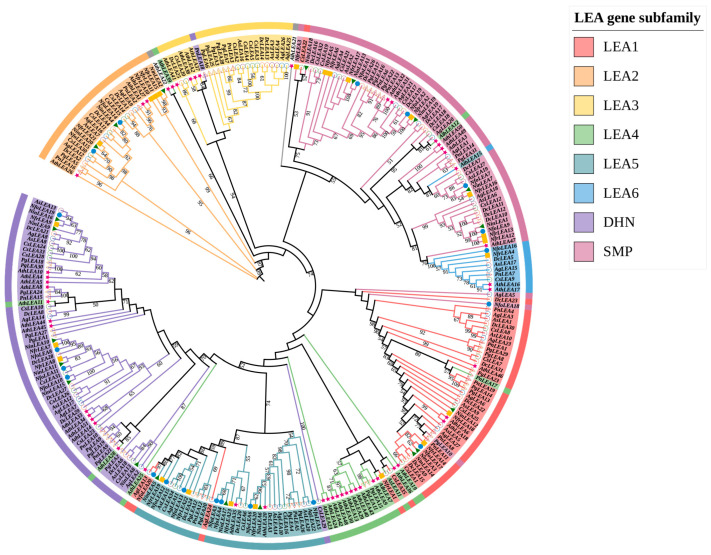
A phylogenetic evolutionary tree of *LEA* gene families in *Notopterygium* species and seven other species. The differently shaped symbols preceding the gene IDs represent different species, with solid symbols denoting the three species of *Notopterygium* and *A. thaliana* (represented by a star), the differently colored hollow circles representing the four species of Apiaceae, and the differently colored hollow triangles representing the two species of Araliaceae.

**Figure 2 ijms-26-00519-f002:**
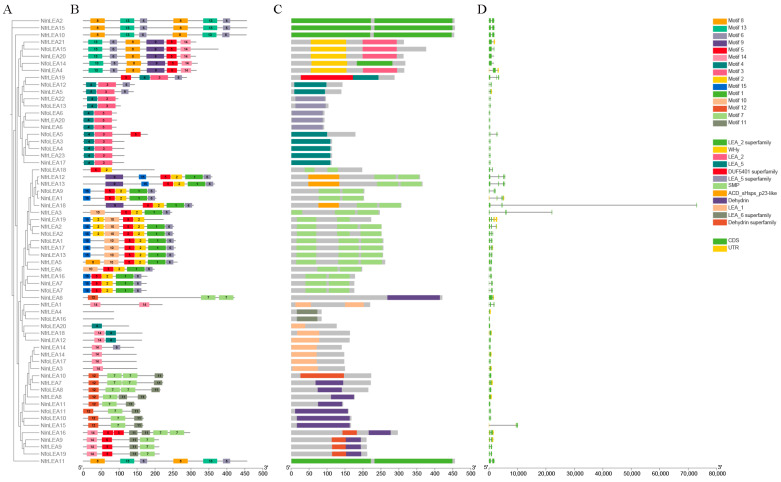
The phylogenetic relationships, distribution of conserved motifs, domain prediction, and gene structure analysis of *Notopterygium LEA* genes. (**A**) A phylogenetic tree of 23 *NfrLEA*, 20 *NfoLEA*, and 20 *NinLEA* genes. (**B**) The distribution of conserved motifs in LEA proteins. Fifteen conservative motifs in LEA are displayed in different color boxes. (**C**) The domains of LEA proteins. These denote the matching types that represent various confidence levels (specific matching and non-specific matching) and domain model ranges (superfamily and multi-domain). (**D**) *LEA* gene structures. The description of exons and introns is obtained by adopting image tools. The yellow and green boxes and the green lines represent the non-coding region (UTR) and exons and introns, respectively.

**Figure 3 ijms-26-00519-f003:**
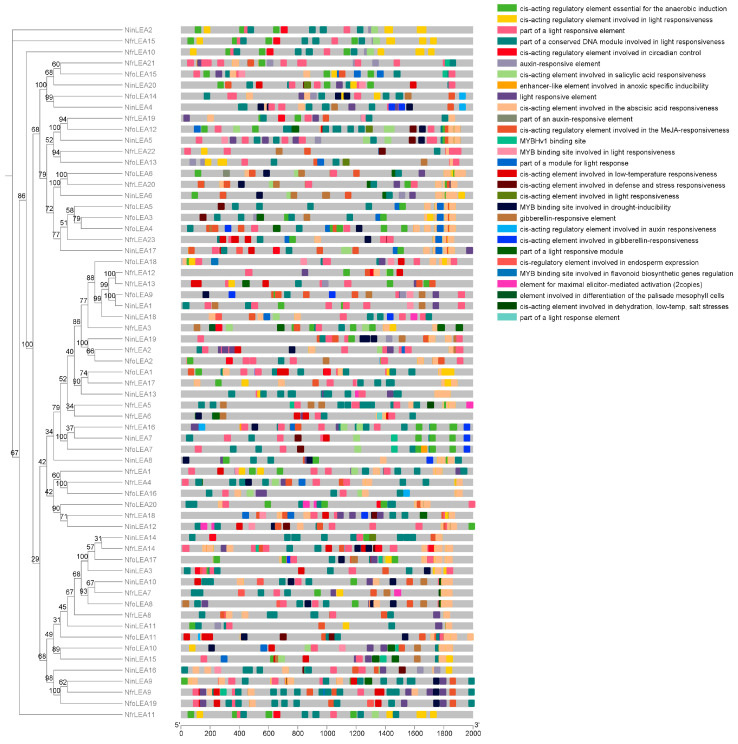
Prediction of *cis*-acting elements in promoter region of *LEA* genes in *Notopterygium* species.

**Figure 4 ijms-26-00519-f004:**
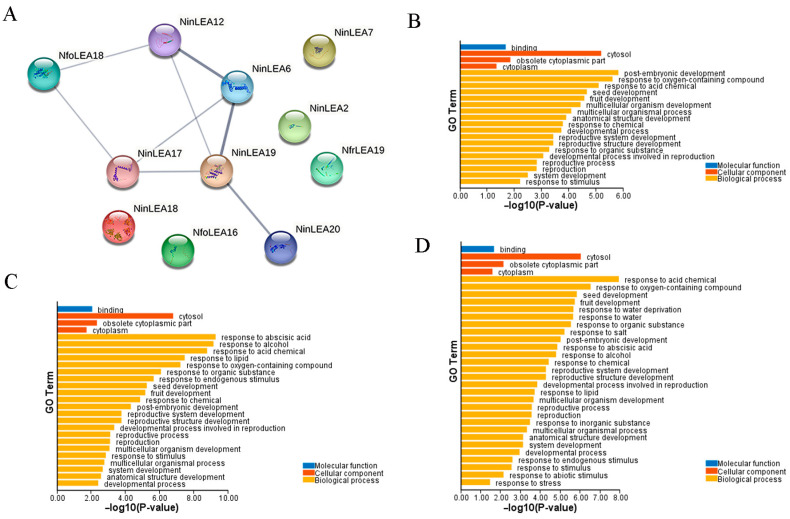
Protein–protein interaction analysis of LEA proteins and result of GO annotation of LEA genes in *Notopterygium* species. (**A**) PPI network in *Notopterygium* species. (**B**) GO functional annotation of *NfrLEAs*. (**C**) GO functional annotation of *NfoLEAs*. (**D**) GO functional annotation of *NinLEAs*.

**Figure 5 ijms-26-00519-f005:**
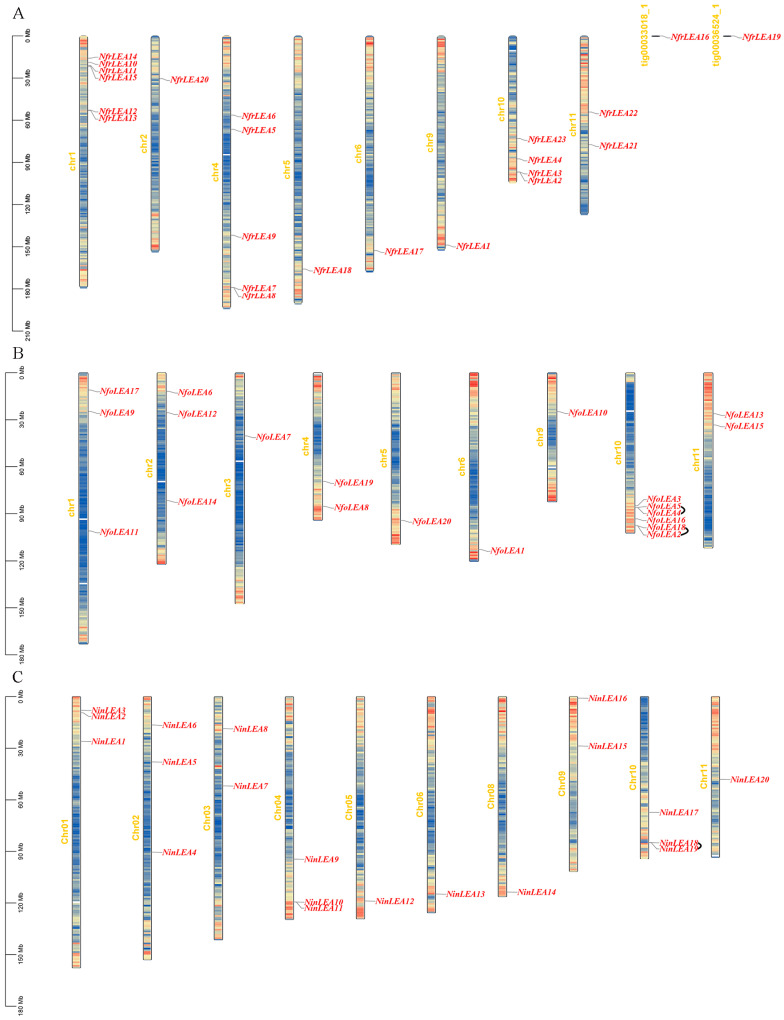
A chromosomal localization analysis of *Notopterygium LEA* genes. (**A**) The localization of *NfrLEAs*. (**B**) The localization of *NfoLEAs*. (**C**) The localization of *NinLEAs*.

**Figure 6 ijms-26-00519-f006:**
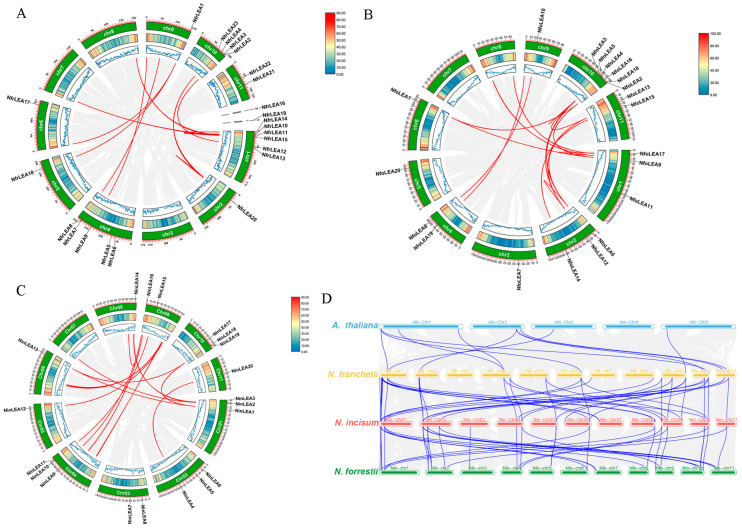
Interchromosomal relationships of *Notopterygium LEA* genes. (**A**) Intra-species collinearity analysis of *NfrLEAs*. (**B**) Intra-species collinearity analysis of *NfoLEAs*. (**C**) Intra-species collinearity analysis of *NinLEAs*. (**D**) Collinearity of *LEA* gene families in *Notopterygium* species. Gray lines indicate all collinearity blocks in three *Notopterygium* genomes, and red and blue lines indicate duplicated *LEA* gene pairs.

**Figure 7 ijms-26-00519-f007:**
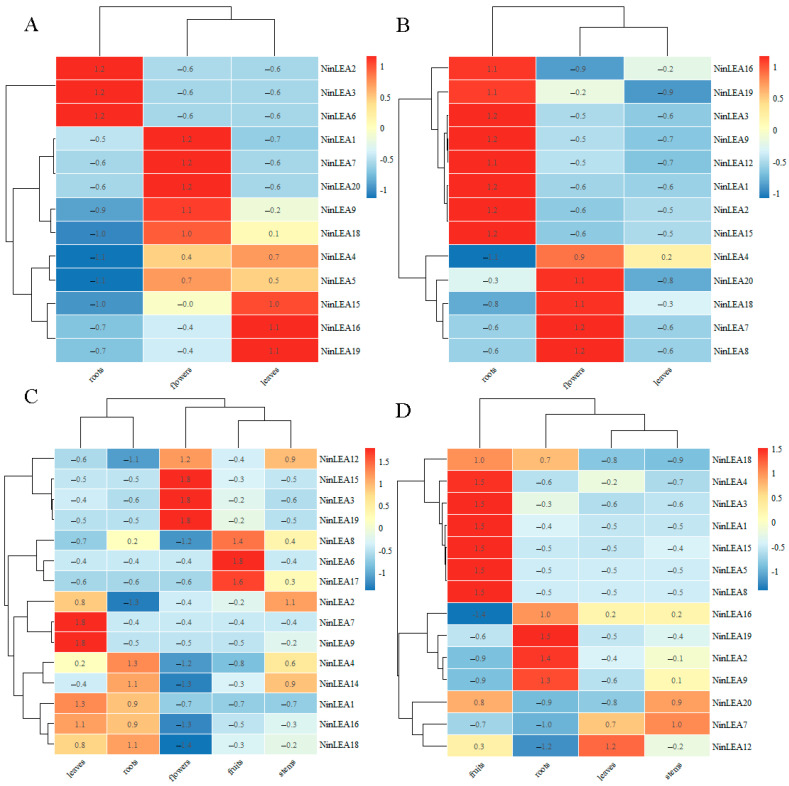
Gene expression heat map of *NinLEAs* in different tissues of four *Notopterygium* species. (**A**) Expression patterns of *NinLEAs* in different tissues of *N. franchetii*. (**B**) Expression patterns of *NinLEAs* in different tissues of *N. incisum*. (**C**) Expression patterns of *NinLEAs* in different tissues of *N. forrestii*. (**D**) Expression patterns of *NinLEAs* in different tissues of *N. oviforme*.

**Figure 8 ijms-26-00519-f008:**
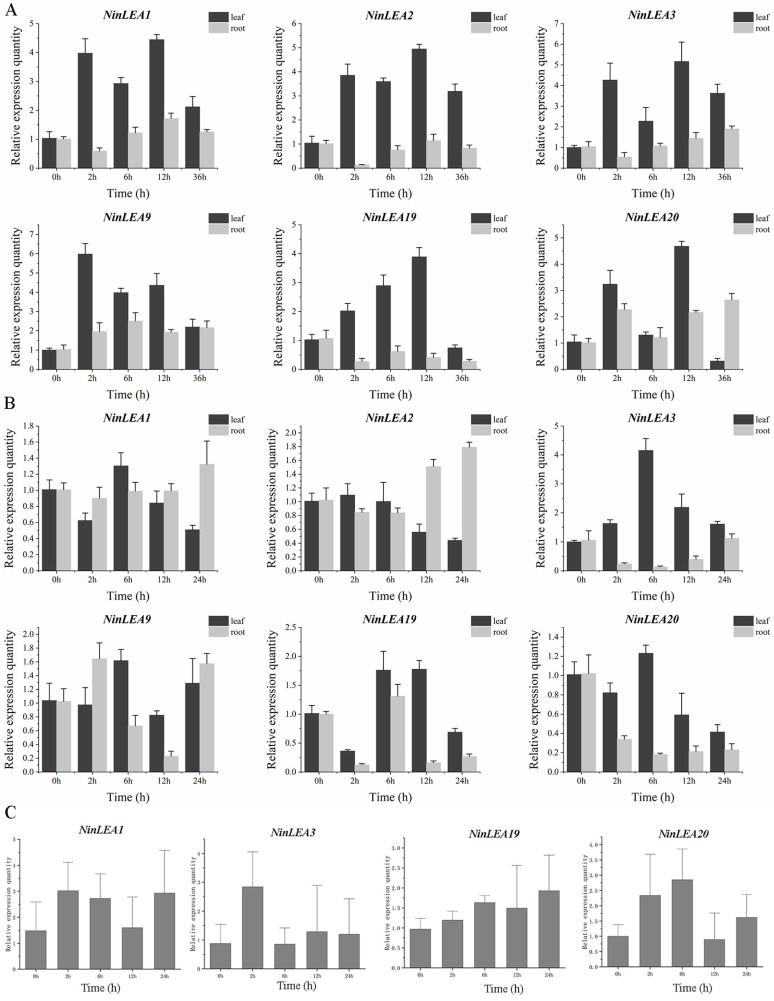
Expression analysis of *NinLEAs* in *N. franchetii* under abiotic stress. (**A**) Relative expression levels of *NinLEAs* in leaves and roots under high-temperature stress. (**B**) Relative expression levels of *NinLEAs* in leaves and roots under drought stress. (**C**) Relative expression levels of *NinLEAs* in leaves under low-temperature stress.

## Data Availability

All data are contained within the article and the Appendix A.

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
