# Peer review of "Genome-Wide Identification, Phylogenetic Evolution, and Abiotic Stress Response Analyses of the Late Embryogenesis Abundant Gene Family in the Alpine Cold-Tolerant Medicinal Notopterygium Species"

_ijms, 2025, doi:10.3390/ijms26020519_

Round 1
Reviewer 1 Report
Comments and Suggestions for Authors The study of the LEA gene family in Notopterygium plants provides theoretical support for further exploring their functions and stress resistance mechanisms in high-altitude cold tolerant plants, laying the foundation for molecular breeding. However, there is a lack of experimental data on cold resistance, and we hope to supplement and improve it.
(The description of cold resistance lacks sufficient evidence and requires additional relevant experiments on the NinLEA gene)
宁LEA 基因
宁LEA 基因
Author Response
Comments 1: [The description of cold resistance lacks sufficient evidence and requires additional relevant experiments on the NinLEA gene]
Response 1: Thank you for pointing this out. We agree with this comment. Therefore, after receiving the feedback, we conducted relevant cold tolerance experiments. The plants were placed in a 0°C environment, and samples were taken at five time points [0h, 2h, 6h, 12h, 24h]. RNA extraction, reverse transcription, and qPCR analysis were performed. Initially, we found that the plant leaves did not show significant changes under continuous low-temperature conditions, which was consistent with the results of our pre-experiment under low-temperature conditions. Furthermore, the relative expression levels of NinLEA1, NinLEA3, NinLEA19 and NinLEA20 exhibited a trend of increasing first and then decreasing. we found that NinLEA3 displayed a similar expression pattern under both high and low temperature conditions. [The revisions in the manuscript are as follows: the relative expression levels of NinLEA genes under low temperature are shown in panel C of Figure 8; Added the following to line 370: "(C) Relative expression levels of NinLEAs in leaves under low temperature stress." The modifications in section 2.7 of the results are in lines 347, 351, and 358-360; the revisions in the discussion section are in lines 511, 516-529; we have reorganized part 4.7 of the materials and methods, with modifications in lines 592-602; the modification in the conclusion section is in line 633.]

Reviewer 2 Report
Comments and Suggestions for Authors
Authors identified 63 Late embryogenesis abundant (LEA) genes from three Notopterygium species. Moreover, Authors studied the gene structure, chromosomal localization, cis-element distribution within promoters as well as aa conserved motifs distribution in corresponding proteins. Furthermore, the phylogenetic and the protein interaction network analysis was presented. Authors evaluated the organ-specific LEAs gene expression based on transcriptomic studies and verified the responsiveness to heat and cold treatment by qRT-PCR analysis.
Results are mosty based on data available in databases. However, Authors added own qRT-PCR results. Obtained information could be interesting to researchers in the field. Study is well planned and performed, conclusions are supported by results. Article is well written. Following comments should be addressed to further improve the manuscript.
1. Lines 94-99. Add the citation of Jia et al. 2017. Line 154 add citation of Hundertmark et al., 2008.
Check the whole text for similar errors.
2. Section 3.3-start the title with capital letter, should be Gene not gene.
3. Figure 2- add A, B, C, D to mark parts of Fig. 2
4. Line 271 should be GO not Go.
5. Check the font size in line 307.
6. Section 2.8. Add following information:
I. Line 572- should be NanoDrop2000
II. Add name, manufacturer and country of origin of qPCR instrument.
III. Provide approximate amount of RNA/cDNA per one qPCR tube/sample.
IV. Add the length of PCR amplicons for tested and reference gene
V. Add citation of previous use of reference gene or provide analysis of gene expression stability using BestKeeper or related software.
VI. From which plant was the reference gene actin8?
VII. Add the name of software used to acquire the raw RT-PCR data
Author Response
Comments 1: [Lines 94-99. Add the citation of Jia et al. 2017. Line 154 add citation of Hundertmark et al., 2008.]
Response 1: Thank you for pointing this out. We agree with this comment. We have completed the necessary revisions and standardized the reference format. The modifications are located on lines 99, 102, and 159 of the manuscript.
Comments 2: [Section 3.3-start the title with capital letter, should be Gene not gene.]
Response 2: Thank you for pointing this out. We agree with this comment. We have completed the necessary revisions and standardized the format of all secondary headings. The modifications are located on lines 175 of the manuscript.
Comments 3: [Figure 2- add A, B, C, D to mark parts of Fig. 2]
Response 3: Thank you for pointing this out. We agree with this comment and apologize for our carelessness in expression. We have completed the necessary revisions. The modifications are located on lines 234 of the manuscript.
Comments 4: [Line 271 should be GO not Go.]
Response 4: Thank you for pointing this out. We agree with this comment. We have completed the necessary revisions and, through review, corrected the numbering of the secondary headings in the Results and Methods sections. The modifications are located on lines 279 of the manuscript.
Comments 5: [Check the font size in line 307.]
Response 5: Thank you for pointing this out. We agree with this comment. We have completed the necessary revisions. The modifications are located on lines 316 of the manuscript.
Comments 6: [Section 2.8. Add following information: I. Line 572- should be NanoDrop2000;II. Add name, manufacturer and country of origin of qPCR instrument;III. Provide approximate amount of RNA/cDNA per one qPCR tube/sample.IV. Add the length of PCR amplicons for tested and reference gene. V. Add citation of previous use of reference gene or provide analysis of gene expression stability using BestKeeper or related software. VI.From which plant was the reference gene actin8? VII. Add the name of software used to acquire the raw RT-PCR data]
Response 6: Thank you for pointing this out. We agree with this comment. We have added the necessary information in this section. The revisions are as follows:
I.Line 572- should be NanoDrop2000. [Done]
II.Add name, manufacturer and country of origin of qPCR instrument. [Done]
III. Provide approximate amount of RNA/cDNA per one qPCR tube/sample. [Done]
IV.Add the length of PCR amplicons for tested and reference gene. [Done]
V.Add citation of previous use of reference gene or provide analysis of gene expression stability using BestKeeper or related software.[Done]
VI.From which plant was the reference gene actin8? [Done]
VII. Add the name of software used to acquire the raw RT-PCR data. [The raw qRT-PCR data is exported from the qRT-PCR device in the form of a table to obtain the raw CT values.] The revisions of the above section are in lines 603-620 of the manuscript.

Round 2
Reviewer 2 Report
Comments and Suggestions for Authors
Authors corrected the manuscript according to reviewer comments. I have no other remarks.